# Scaling Embedding Layers in Language Models

## Abstract

We propose SCONE (**S**calable, **C**ontextualized, **O**ffloaded, **N**-gram **E**mbedding), a method for extending input embedding layers to enhance language model performance as layer size scales. To avoid increased decoding costs, SCONE retains the original vocabulary while introducing embeddings for a set of frequent $n$-grams. These embeddings provide contextualized representation for each input token and are learned with a separate model during training. During inference, they are precomputed and stored in off-accelerator memory with minimal impact on inference speed. SCONE enables two new scaling strategies: increasing the number of cached $n$-gram embeddings and scaling the model used to learn them, all while maintaining fixed inference-time FLOPS. We show that scaling both aspects allows SCONE to outperform a 1.9B parameter baseline across diverse corpora, while using only half the inference-time FLOPS.

## 1. Introduction

Embedding layers in language models map discrete tokens to continuous vector representations (Mikolov, 2013; Sennrich et al., 2016). These layers can be implemented as lookup tables, enabling efficient retrieval of embeddings using hash- or tree-based data structures. This allows embedding layers to be offloaded to main memory or even secondary storage (e.g., disk) with minimal impact on inference speed. This is desirable, as main memory and secondary storage are significantly more cost-effective than accelerators (e.g., GPUs and TPUs (McCallum, 2024)). These advantages drive our exploration of methods for scaling up embedding layers.

However, scaling the embedding layer by simply increasing the vocabulary size has limited benefits. The first issue is the

---

[1]Anonymous Institution, Anonymous City, Anonymous Region, Anonymous Country. Correspondence to: Anonymous Author <anon.email@domain.com>.

Preliminary work. Under review by the International Conference on Machine Learning (ICML). Do not distribute.

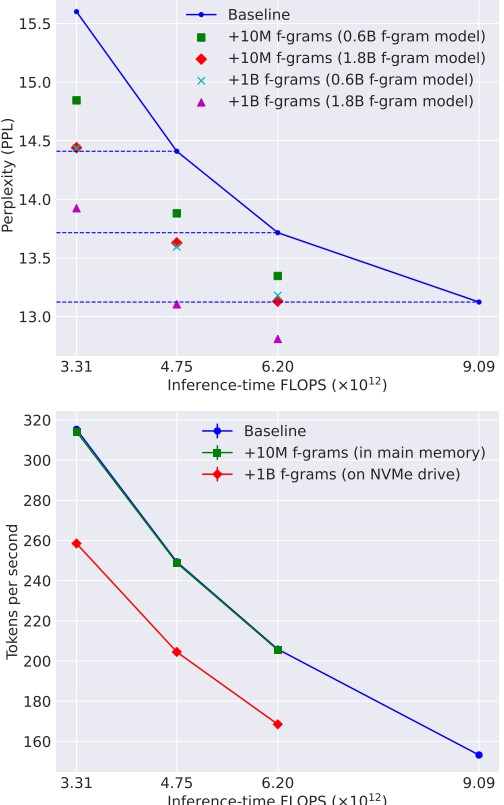

*Figure 1.* (**Top**) Perplexity (lower is better) on the OLMo (Groeneveld et al., 2024) evaluation mixture. Inference-time FLOPS refer to the forward pass computation cost for four model sizes (0.7B, 1B, 1.3B, and 1.9B). With 10M f-grams, the 1.3B model matches the 1.9B baseline, while with 1B f-grams, the 1B model surpasses it. (**Bottom**) End-to-end token generation speed on a single A100 using vLLM (Kwon et al., 2023). Storing f-gram embeddings in main memory introduces negligible latency, while NVMe storage slows generation slightly but does not create a bottleneck.

coupling between the input embedding layer and the output (logits) layer: (i) It is common to share weights between the input embedding and the output layer. In this case, the weights already reside in the accelerator memory as they are needed for logits computation, which eliminates the benefits of offloading the input embedding. (ii) Even when the weight *parameters* are not shared, the weight *shapes* are tied to the vocabulary size and embedding dimension. When scaling the input embedding by increasing the vocabulary

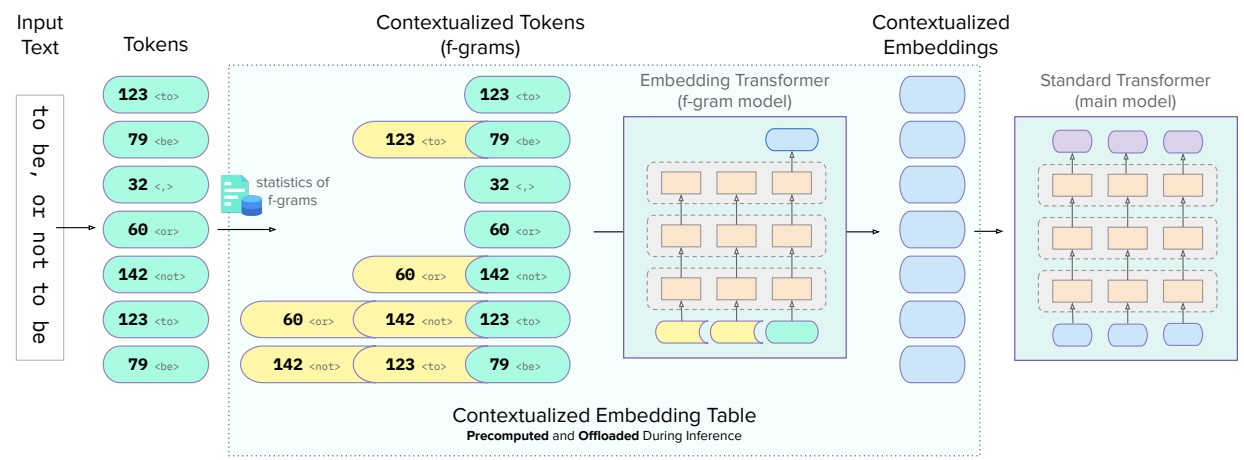

*Figure 2.* Illustration of SCONE (with a maximum $n$-gram length of 3). The term *f-grams* refers to the set of frequent $n$-grams (Section 3).

size, as explored in prior studies (Wang et al., 2019; Zheng et al., 2021; Liang et al., 2023; Tao et al., 2024), the inference cost also increases due to the size growth of the output embedding that is used to compute logits (Dagan et al., 2024). This scaling becomes computationally impractical beyond a vocabulary size of a few hundred thousands.

The second issue is that the benefits of scaling the vocabulary diminish, even when computational costs can be mitigated by advances in accelerators or smarter algorithms (Joulin et al., 2017; Shim et al., 2017): Scaling leads to a large number of *tail tokens*, that have low frequency in the training corpus. The training of their (input and output) embeddings receives very few updates which results in representations that are of lower quality (Liao et al., 2021; Dou et al., 2024). Our experiments with GPT-2 models (Radford et al., 2019) pre-trained on WebText (Peterson et al., 2019) confirm these limitations: Only 7.3% of embedding vectors in a 2M vocabulary receive more than 100 updates over 100M training tokens, compared to 97.6% for a 32K vocabulary. Additionally, we observe performance degradation and a linear increase in accelerator memory usage when the vocabulary size exceeds 1M (Appendix C).

**Our contributions.** In this paper we propose a novel approach to disentangle the input and output embeddings, bypassing those issues and enabling the effective input embedding scaling with minimal additional inference cost. Instead of increasing the size of the vocabulary, we augment each token in the base vocabulary with $n$-gram contextualized variants. The $n$-grams are selected from a predefined set of frequently occurring $n$-grams, that we refer to as *f-grams*. Those contextualized tokens are only used in input embedding computation, allowing us to build an augmented input embedding table with billions of entries without impacting the computation cost of the output layer. Furthermore, the embeddings for those contextualized tokens are generated

from an embedding transformer model, referred to as *f-gram model*, that is jointly trained (see Figure 2 for an illustration). This allows us to obtain rich contextualized representations without being subjected to the sparse tail phenomenon of naively increasing the vocabulary size.

With this novel design, we introduce two new directions for improving model performance: (i) increasing the number of cached f-gram embeddings and (ii) scaling up the f-gram model for learning those embeddings. Notably, both approaches preserve inference-time FLOPs. These directions enable us to fully leverage the precomputation and offloading of contextualized embeddings. We demonstrate that a 1B parameter model with SCONE outperforms a baseline model requiring $\sim 2\times$ more inference-time FLOPs. Figure 1 presents representative results from Section 4.2.

To summarize, our contributions are as follows:
- We propose SCONE, a scalable approach to expand the embedding layer in language models (Section 3).
- We conduct extensive experiments to evaluate design choices and validate our method in large-scale pre-training setups (Section 4).

## 2. Preliminaries

We focus on pre-training decoder-only language models using the causal language modeling objective, a standard recipe to train modern language models (Radford et al., 2019; Brown et al., 2020). This involves predicting the next token based solely on preceding tokens, enabling these models to handle diverse text generation tasks.

We introduce formal notations to describe such model architectures. Following Phuong & Hutter (2022), we prioritize clarity and omit details that are not essential for describing our method. We assume that a vocabulary $V_{\text{token}}$ of tokens

has already been defined. The *token embedding layer* is parameterized by a function $\mathcal{T} : V_{\text{token}} \to \mathbb{R}^d$ that maps any token in $V_{\text{token}}$ to an embedding vector in $\mathbb{R}^d$, where $d$ is the embedding dimension. We abstractly view a transformer model as $\mathcal{A} : (\mathbb{R}^d)^{\leq N_{\max}} \to \mathbb{R}^d$, which maps a sequence of vectors in $\mathbb{R}^d$ of length at most $N_{\max}$ to a single embedding vector[1] in $\mathbb{R}^d$. The size of a transformer model refers to the number of parameters in the model. A prediction head $\mathcal{D} : \mathbb{R}^d \to \Delta_{V_{\text{token}}}$ maps the embedding vector to a probability distribution over tokens in the vocabulary $V_{\text{token}}$ (where $\Delta_{V_{\text{token}}}$ denotes the probability simplex). Collectively these pieces yield a basic next-word prediction model $M_{\mathcal{T},\mathcal{A},\mathcal{D}}$, which, given an input sequence $(\sigma_1, \ldots, \sigma_m) \in V_{\text{token}}^m$ of tokens, produces a distribution over the next token $\hat{\sigma}_{m+1}$, which can be used for auto-regressive sequence prediction. For completeness, we present the pesudocode of the basic next-word prediction model in Algorithm 3 (Appendix B).

*Remarks on Token Embedding Layers.* The number of parameters in a token embedding layer $\mathcal{T} : V_{\text{token}} \to \mathbb{R}^d$ is the product of the vocabulary size $|V_{\text{token}}|$, which ranges from a few dozen to hundreds of thousands in current models (Radford et al., 2019; Dubey et al., 2024), and the embedding dimension $d$, which is typically ranges in a few thousands (Brown et al., 2020; Touvron et al., 2023). Often, the prediction head $\mathcal{D} : \mathbb{R}^d \to \Delta_{V_{\text{token}}}$ computes the similarity between the model's output embedding and all vocabulary embeddings to predict the next token, thereby sharing the embeddings in the token embedding layer.

Embedding layers admit highly efficient implementations. The key-value pairs can be organized with hash- or tree-based data structures, which allow the embedding layer to operate with access cost that is either constant or logarithmic in the number of embedding vectors. These low cost methods, in principle, allow the embedding layer to be offloaded from accelerators with minimal latency impact (see Section 3.3). Most implementations, however, store the token embedding layer in accelerator memory, since the token embeddings are also required for applying the prediction head.

## 3. SCONE Architecture

We introduce the SCONE architecture that uses a new embedding layer for frequently occurring $n$-grams of tokens. Figure 2 shows the high-level approach.

First we construct a set $V_{\text{f-gram}} \subseteq V_{\text{token}}^{[2,n]} := \bigcup_{k=2}^{n} V_{\text{token}}^k$ of frequently occurring $k$-grams of length at most $n$, that we term *f-grams*; throughout this paper, we use $n$ to denote

---

[1] The output embedding vector is typically the last token embedding, which is commonly used for next-token prediction. In this work, for the f-gram model, we use the last token embedding as the contextualized embedding of an input token.

---

**Algorithm 1** Constructing a set of f-grams $V_{\text{f-gram}}$.

**Parameters:** $S$: desired size of $V_{\text{f-gram}}$.
**Input:** $\{(\sigma_1, \ldots, \sigma_{N_{\max}})^{(i)}\}$ : token sequences from training set, where each sequence is from $V_{\text{token}}^{N_{\max}}$.
**Output:** $V_{\text{f-gram}} \subseteq V_{\text{token}}^{[2,n]}$: set of f-grams of size $S$.
**for** $k = 2, \ldots, n$ **do**
    **for** $\omega := (\sigma_1', \ldots, \sigma_k') \in V_{\text{token}}^k$ **do**
        $C_\omega \leftarrow$ the number of times $\omega$ appears in all sequences $\{(\sigma_1, \ldots, \sigma_{N_{\max}})^{(i)}\}$.
Let $\omega_1, \omega_2, \ldots$ be list of elements of $\bigcup_{k=2}^{n} V_{\text{token}}^k$, sorted such that $C_{\omega_1} \geq C_{\omega_2} \geq \cdots$, breaking ties arbitrarily.
**return** $\{\omega_1, \ldots, \omega_S\}$: set of f-grams of size $S$

---

**Algorithm 2** SCONE method $F_{\mathcal{T}, V_{\text{f-gram}}, \mathcal{A}_{\text{f-gram}} | \mathcal{F}}$.

**Parameters:**
- $\mathcal{T} : V_{\text{token}} \to \mathbb{R}^d$: token embedding layer,
- $V_{\text{f-gram}} \subseteq V_{\text{token}}^{\leq n}$: set of f-grams,
- **TRAINING**: f-gram transformer model
    ▷ $\mathcal{A}_{\text{f-gram}} : (\mathbb{R}^d)^{\leq n} \to \mathbb{R}^d$,
- **INFERENCE**: f-gram embedding layer
    ▷ $\mathcal{F} : V_{\text{f-gram}} \to \mathbb{R}^d$.

**Input:** A sequence of $m$ tokens $(\sigma_1, \ldots, \sigma_m) \in V_{\text{token}}^m$.
**Output:** Embeddings $(\boldsymbol{e}_1, \ldots, \boldsymbol{e}_m) \in (\mathbb{R}^d)^m$.
**for** $i = 1, 2, \ldots, m$ **do**
    $j \leftarrow$ smallest $j' < i$ s.t. $(\sigma_{j'}, \ldots, \sigma_i) \in V_{\text{f-gram}}$ if such a $j'$ exists, otherwise $i$.
    **if** $j = i$ **then**
        $\boldsymbol{e}_i \leftarrow \mathcal{T}(\sigma_i)$.
    **else**
$$\boldsymbol{e}_i \leftarrow \begin{cases} \mathcal{A}_{\text{f-gram}}(\mathcal{T}(\sigma_j), \ldots, \mathcal{T}(\sigma_i)) & \text{at training} \\ \mathcal{F}(\sigma_j, \ldots, \sigma_i) & \text{at inference} \end{cases}$$
**return** $(\boldsymbol{e}_1, \ldots, \boldsymbol{e}_m)$

---

the maximum length of f-grams considered. Algorithm 1 illustrates this process. That said, Algorithm 1 is for illustration purposes only; in practice, we use a more efficient implementation requiring only $(n - 1)$ linear scans over the training corpus, as detailed in Section 3.1. Notably, the counting and ranking process resembles continuing the training of a BPE tokenizer (Sennrich et al., 2016) with the existing vocabulary.

Next, we define the SCONE method, which maps a given sequence of tokens to a sequence of embedding vectors. SCONE method behaves differently at training and at inference, as described in Algorithm 2. At training, it is parameterized by an *f-gram transformer model* $\mathcal{A}_{\text{f-gram}}$. However, at inference, it is parameterized instead by an *f-gram embedding layer* $\mathcal{F} : V_{\text{f-gram}} \to \mathbb{R}^d$ that maps the set of f-grams obtained above to embedding vectors. This embedding layer is implemented by caching the outputs of $\mathcal{A}_{\text{f-gram}}$ for all f-grams in $V_{\text{f-gram}}$ and storing them in a hash- or tree-based

data structure for efficient retrieval.

The embeddings produced by SCONE are then passed to a standard transformer model $\mathcal{A}_{\text{main}} : (\mathbb{R}^d)^{\leq N_{\text{max}}} \rightarrow \mathbb{R}^d$, referred to as the *main model*, followed by a prediction head $\mathcal{D} : \mathbb{R}^d \rightarrow \Delta_{V_{\text{token}}}$. Together, these components form the end-to-end process for next-word prediction with SCONE. We present the corresponding pseudocode in Algorithm 4 in Appendix B.

In the rest of this section, we will discuss the motivations behind these design decisions and provide further implementation details.

### 3.1. BPE-Style Discovery of f-grams

The construction of $V_{\text{f-gram}}$, as defined in Algorithm 1, can be implemented efficiently with $n - 1$ linear scans over the training corpus. We perform one scan for each $k \in [2, n]$, starting with 2-grams. In subsequent scans, we impose a minimum frequency threshold of 5 to reduce memory usage. At the $(k+1)_{th}$ scan, the set of $k$-grams from the previous scan allows us to skip any $(k+1)$-gram candidates that cannot meet the threshold. Specifically, if an $(k+1)$-gram surpasses the threshold, its $k$-suffix or prefix must appear at least as many times. Figure 3 shows how the number of unique $k$-grams (up to 6-grams) grows as the training corpus scales from a few billion to one trillion tokens. Finally, all found $k$-grams (for $k \in [2, n]$) are ranked by frequency, and the top $S$ are selected as keys, where $S = |V_{\text{f-gram}}|$ is the target number of f-grams.

Our procedure for counting and ranking the $n$-grams is analogous to continuing a BPE tokenizer's training on an existing vocabulary. In each BPE iteration (Gage, 1994; Sennrich et al., 2016), the frequencies of all token pairs (2-grams) are counted and the most frequent pair is merged to form a new token, expanding the vocabulary by one. However, merging and recounting pairs repeatedly is prohibitively expensive for large corpora. Instead, we simply collect and sort all $n$-grams up to a small $n$.

### 3.2. Learning f-gram Embeddings with $\mathcal{A}_{\text{f-gram}}$

We motivate our use of $\mathcal{A}_{\text{f-gram}}$ by considering the alternative of directly backpropagating gradients to a large embedding table. The issue with the alternative approach is that it does not exploit the dependencies between $n$-grams and therefore suffers from fewer updates per embedding. We explored this by pre-training GPT-2 models with token vocabulary sizes ranging from 32K to 2M. As vocabulary size increases, token embedding updates become sparser, which eventually degrades performance. For example, when training over 100M tokens, 97.6% of the tokens in a 32K vocabulary receive more than 100 updates. In contrast, with a vocabulary of 2M, only 7.3% of the tokens reach this

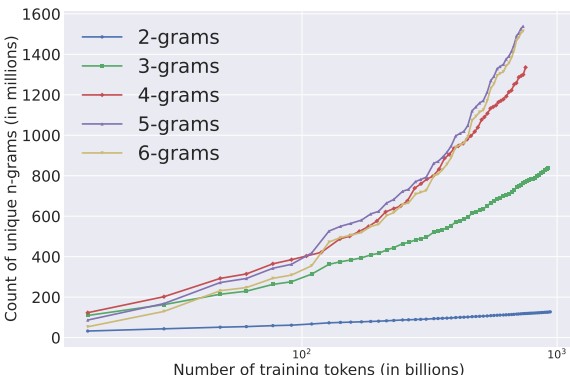

*Figure 3.* Number of unique 2- to 6-grams that appear at least five times. We vary the size of the corpus by uniformly sampling sequences from the OLMo tokenized training corpus (Soldaini et al., 2024).

threshold. See Appendix C for more details. The sparsity makes it extremely challenging to learn an embedding table by directly backpropagating gradients to the embeddings. SCONE solve this problem by parameterizing the embeddings with a f-gram transformer $\mathcal{A}_{\text{f-gram}}$, avoiding the sparse update issue.

SCONE jointly trains the $\mathcal{A}_{\text{f-gram}}$ model with the main model $\mathcal{A}_{\text{main}}$ and the token embedding layer $\mathcal{T}$. This overcomes the sparse updates issue but also introduces additional compute costs. For each $\omega \in V_{\text{f-gram}}$, the computation is the same as that of processing a short sequence of length $|\omega|$ through a standard transformer. Since $|\omega|$ is small ($|\omega| \leq 5$ for most of our experiments), the primary overhead comes from the feed-forward layer.

During inference, the f-gram embedding layer $\mathcal{F}$ can be precomputed and stored in a lookup table, offloaded to CPU memory or secondary storage for efficient retrieval. Meanwhile, the token embedding layer $\mathcal{T}$ remains on the accelerator for decoding. This design leverages the low complexity of embedding layers to enrich token representations without increasing decoding costs.

Importantly, SCONE introduces a novel approach to improving performance under a fixed inference-time FLOPS budget. Prior work shows that increasing training compute beyond a compute-optimal threshold yields diminishing returns for a fixed model size (Hoffmann et al., 2022). A common strategy to utilize extra training compute is scaling up both model size and compute, but this typically raises inference-time FLOPS as well. In contrast, our method allows the $\mathcal{A}_{\text{f-gram}}$ model to benefit from greater training compute without increasing inference-time FLOPS.

### 3.3. Space Usage and Query Latency

We evaluate the space usage and query latency of the f-gram embedding layer under various configurations. We show that latency is not a bottleneck for language model inference and the space costs are low due to the use of relatively inexpensive system memory or secondary storage.

Using the setup described in Section 4.2, we set the maximum $n$-gram length to 5 and experiment with $|V_{\text{f-gram}}|$ being 10M, 100M, and 1B with embedding dimension of $d = 2048$ with 16-bit precision per floating point value. Experiments were conducted on a workstation with 64 Intel Xeon CPU cores and 512 GB of memory. Space and latency were measured for both in-memory and on-disk storage. In memory, embeddings are stored as a single matrix with a hash dictionary mapping f-grams to indices, while on-disk storage uses the Lightning Memory-Mapped Database (Chu, 2011) to directly store f-gram and embedding pairs on NVMe solid-state drives.

Table 1. Space usage of the f-gram embedding layer $\mathcal{F}$, along with cost for memory and NVMe solid-state drives (McCallum, 2024).

| # of $n$-grams | System memory | Solid-state drive |
|---|---|---|
| $10^7$ | 41.4 GB | 77.3 GB |
| $10^8$ | 413.6 GB | 766.8 GB |
| $10^9$ | (does not fit) | 7665.4 GB |
| Price (per GB) | $\sim 2$ USD | $\sim 0.1$ USD |

Table 1 summarizes the space usage for both storage methods. In both cases, the space required increases linearly with the number of embedding vectors. The 10M and 100M f-gram embedding layers are able to fit within CPU memory, with the 10M layer requiring 41.4 GB. For on-disk storage, there is additional overhead as the same 10M layer occupies 77.3 GB storage.

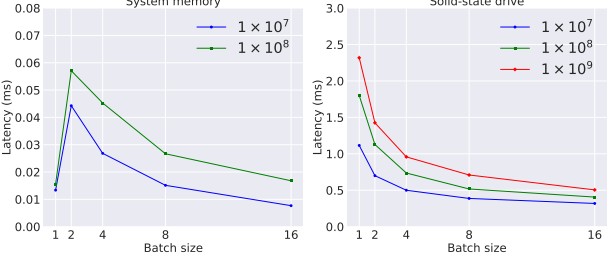

Figure 4. Amortized per-token query latency (ms), averaged over 100,000 batches. The latency spike from batch size 1 to 2 when reading from system memory is due to batch operator overhead, which is less pronounced for solid-state drives.

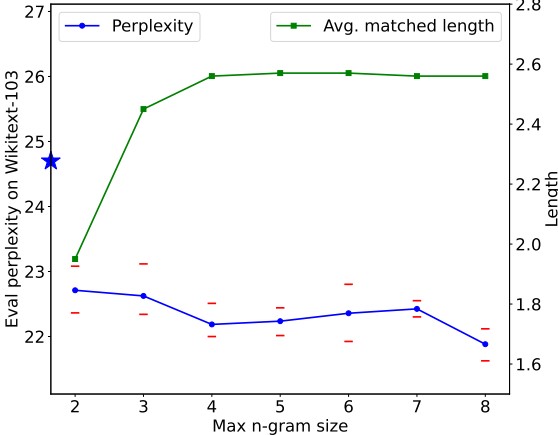

Figure 5. Effect of the maximum f-gram length $n$ in $V_{\text{f-gram}}$, on perplexity and matched length. The left $y$-axis shows perplexity (averaged over three seeds), where the leftmost star indicates baseline performance. The right $y$-axis shows the average length of matched f-grams. The perplexity decreases as we increase the maximum length from 2 to 4, but then plateaus with some fluctuation. Similarly, the average matched length initially rises but stabilizes after size 4.

Figure 4 shows the latency of retrieving embeddings with different batch sizes. Latency is measured as the end-to-end time from loading a batch of tokens to the f-gram embeddings (Algorithm 2) being ready on GPU. CPU cache is cleared before each test, and up to 4 queries are made per token to identify the longest matching $n$-gram (with a maximum length of 5). For in-memory storage, sequential queries suffice as they are not the bottleneck. In contrast, for on-disk storage, we make parallel queries to the database. At a batch size of 1, the latency for a 10M f-gram embedding layer on the NVMe drive is 1.1ms, increasing to 2.3ms for a 1B f-gram embedding layer. This is well below the latency threshold for LLM inference, as typical commercial APIs offer a generation speed of $\sim$100 tokens per second, corresponding to $\sim$10ms per token (ArtificialAnlys, 2025). Larger batch sizes further improve efficiency, with a batch size of 16 reducing the amortized per-token latency to 0.5ms. In-memory access is much faster: for a 100M f-gram embedding layer and a batch size of 16, the amortized per-token latency is only 0.017ms.

## 4. Experimental Evaluation

### 4.1. Design Choices

We analyze three key hyperparameters: (i) the maximum f-gram length in $V_{\text{f-gram}}$, (ii) the number of f-grams used $|V_{\text{f-gram}}|$, and (iii) the $\mathcal{A}_{\text{f-gram}}$ model size. We use the released GPT-2 tokenizer, which has $|V_{\text{token}}| = 50{,}257$, and train on the WebText dataset (Peterson et al., 2019). The tokenized corpus contains 9B training tokens, from which

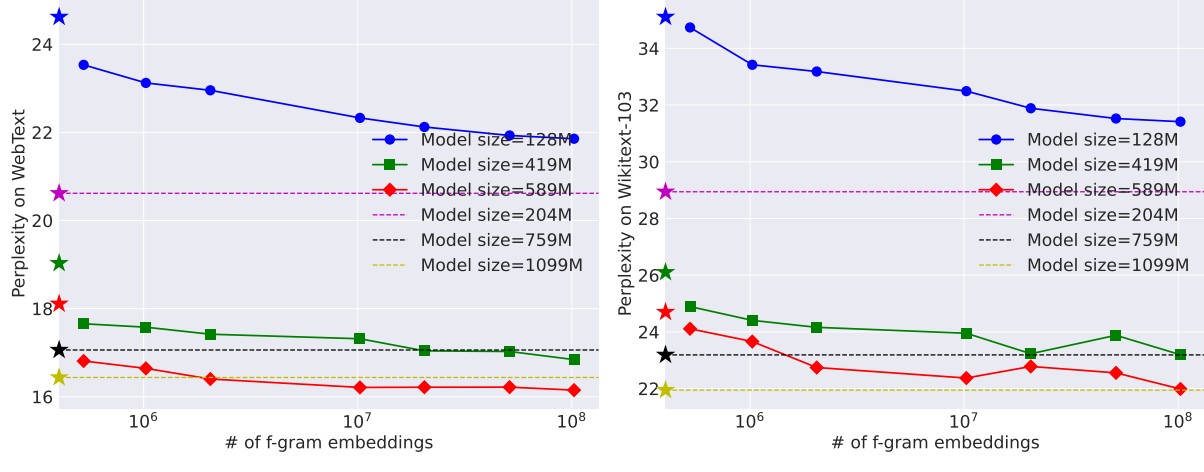

*Figure 6.* Evaluation perplexity as a function of $|V_{\text{f-gram}}|$. Model sizes in the legend correspond to the main model sizes, including the token embedding layer. The dashed lines and leftmost stars indicate baseline performance. Perplexity decreases overall with increasing sizes of $V_{\text{f-gram}}$.

we extract f-grams using the method in Section 3.1.

We consider three main model sizes with 76M, 340M, and 510M non-embedding parameters. Including the token embedding layer, the total parameter count increases to 128M, 419M, and 589M, respectively. The embedding dimensions for these models are 1024, 1536, and 1536, respectively. These models are either trained using only the token embedding layer as baselines or with an additional $\mathcal{A}_{\text{f-gram}}$ when SCONE is applied. Following Radford et al. (2019), we use a batch size of 512 and a sequence length of 1024. Since Radford et al. (2019) do not specify the number of training steps, we train all models for 80B tokens, roughly twice the number of training tokens in Radford et al. (2018). For evaluation, we use the validation split of WebText and WikiText-103 (Merity et al., 2016), one of the largest downstream datasets in Radford et al. (2019). Additional implementation details are in Appendix E.1.

### 4.1.1. VARYING THE MAXIMUM F-GRAM LENGTH

We explore the effect of varying the maximum length $n$ of f-grams in $V_{\text{f-gram}}$. We vary $n$ from 2 to 8 while fixing the total number of f-grams to $|V_{\text{f-gram}}|$ =20M. This means that we obtain a *frequency cutoff* (the minimum frequency of an $n$-gram in $V_{\text{f-gram}}$) that increases with $n$: This value was 7 for $n = 2$ and 108 for $n = 8$. We then measure for each $n$ (i) evaluation perplexity and (ii) the average length of a matched f-grams on Wikitext-103. Our findings are reported in Figure 5. We observe that evaluation perplexity increases for $n$ between 2 and 4 and then plateaus with some fluctuations. A similar trend is observed for the *average match length*, which is the average length of f-gram found in SCONE method (Algorithm 2) for each token in the evalua-

tion set: it rises from 2 to 4 before stabilizing. This is likely because longer f-grams occur less frequently than shorter ones after ranking. Even with a higher maximum $n$-gram length, most selected entries remain short. Additionally, longer f-grams from the training corpus are less likely to match downstream data. Findings on the validation split of WebText (Appendix D.1) exhibit a similar pattern, though the average matched length plateaus later, at length 6.

Considering these findings, for the experiments in the remainder of this paper, we set the maximum f-gram length to $n = 5$ unless stated otherwise.

### 4.1.2. VARYING THE NUMBER OF F-GRAMS

We observe consistent improvements in language modeling performance as we scale up $|V_{\text{f-gram}}|$. To implement the f-gram model $\mathcal{A}_{\text{f-gram}}$, we replicate the baseline model architecture but remove the token embedding layer. This results in the size of $\mathcal{A}_{\text{f-gram}}$ matches the baseline model's non-embedding parameters.

Figure 6 shows the evaluation perplexity as $|V_{\text{f-gram}}|$, the number of f-gram embeddings, increases from 512K to 100M. On the WebText validation split, the perplexity decreases consistently as the number of f-gram embeddings increases. Similarly, on WikiText-103, the perplexity generally decreases with more f-gram embeddings, though minor fluctuations are observed.

In Figure 6, we include three additional baselines where the non-embedding parameters of the three main models are doubled, resulting in models with 204M, 759M, and 1099M parameters for the original 128M, 419M, and 589M models, respectively. This ensures that the total parameter

count of each baseline matches the training-time parameter count when SCONE is applied. With 100M f-gram embeddings, the 419M and 589M models trained with SCONE match or surpass the performance of the 759M and 1099M baselines, respectively, despite using only half as many non-embedding parameters during inference.

### 4.1.3. VARYING THE SIZE OF THE $\mathcal{A}_{\text{f-gram}}$ MODEL

*Figure 7.* Evaluation perplexity on Wikitext-103 as a function of the size of $\mathcal{A}_{\text{f-gram}}$. Model sizes in the legend correspond to the main model sizes, including the token embedding layer. Dashed lines and stars on the left represent baseline performance. The perplexity improves as the size of $\mathcal{A}_{\text{f-gram}}$ grows.

We observe that, for a fixed $|V_{\text{f-gram}}|$, scaling up the $\mathcal{A}_{\text{f-gram}}$ model size provides a new way to improve language modeling performance. We vary the model size by changing the number of layers in the main model architecture. For each $\mathcal{A}_{\text{main}}$ model size, we evaluate four $\mathcal{A}_{\text{f-gram}}$ model sizes: 0.5x, 1x, 2x, and 3x the non-embedding parameters of the main model. We set $|V_{\text{f-gram}}|$ to be 100M. Figure 7 presents the evaluation perplexity on Wikitext-103. The observations on WebText validation split are similar, and we present the results in Appendix D.1.

The results in Figure 7 show that the perplexity generally decreases as the $\mathcal{A}_{\text{f-gram}}$ model size increases, although the improvements become smaller as the model size grows larger. For instance, with the 419M main model, a 170M $\mathcal{A}_{\text{f-gram}}$ model improves the perplexity from 26.1 to 23.4, outperforming the 589M baseline (24.7) by a clear margin. Further scaling of the $\mathcal{A}_{\text{f-gram}}$ model to 1020M (resulting in 1439M total parameters during training) lowers the perplexity to 22.1, which is slightly higher than the 1099M baseline (21.9). This suggests that scaling up the $\mathcal{A}_{\text{f-gram}}$ model size initially provides a better scaling curve, but beyond a certain size, it yields a less optimal scaling curve compared to di-

rectly scaling up $\mathcal{A}_{\text{main}}$. However, the latter also increases inference-time FLOPS, whereas scaling $\mathcal{A}_{\text{f-gram}}$ does not, as it is replaced with an off-accelerator lookup table during inference. This highlights our method as a novel way to leverage additional training compute while maintaining fixed inference-time compute.

### 4.2. Scaling Up the Training Corpus

We demonstrate that the two new scaling aspects of SCONE, namely size of $|V_{\text{f-gram}}|$ and parameters in $\mathcal{A}_{\text{f-gram}}$, apply to large-scale pre-training. We use the tokenized training corpus of OLMo (Groeneveld et al., 2024). OLMo uses a BPE-trained tokenizer with $|V_{\text{token}}| = 50,280$. From this corpus, we uniformly sample one trillion tokens to build $V_{\text{f-gram}}$. We evaluate two sizes of $V_{\text{f-gram}}$: 10M and 1B. The cutoff frequencies are 21,956 for 10M f-grams and 70 for 1B f-grams.

We use a base architecture with 18 decoder blocks and 1B parameters. Following the OLMo-1B architecture, we set a context length $N_{\max} = 2048$ and embedding dimension $d = 2048$. To explore parameter scaling, we vary the number of layers, creating four model variants: 0.7B, 1.0B, 1.3B, and 1.9B. SCONE is evaluated on the first three, while the 1.9B model serves solely as a baseline. We test SCONE with two sizes of $\mathcal{A}_{\text{f-gram}}$, 0.6B and 1.8B, matching the non-embedding parameters of the 0.7B and 1.9B variants. All models are trained on 200B tokens, with sequences of tokens sampled uniformly from the corpus. While Hoffmann et al. (2022) suggests $\sim$20B tokens for a compute-optimal 1B model, we use a larger corpus to approach convergence. Training loss curves are provided in Appendix D.2, with implementation details in Appendix E.2.

Figure 1 presents the perplexity on the OLMo evaluation mixture[2], which comprises 11 diverse corpora, including web crawl data, literature, online forums, scientific writing, coding, and more. Table 2 presents the performance breakdown for the 1B, 1.3B, and 1.9B variants. In Figure 1, the X-axis represents inference-time forward FLOPS per sequence, computed following the breakdown in Hoffmann et al. (2022). Results indicate that increasing both $|V_{\text{f-gram}}|$ and the size of $\mathcal{A}_{\text{f-gram}}$ consistently improves performance across all evaluation corpora. Additionally, in Figure 1 we report end-to-end token generation speed using the vLLM framework (Kwon et al., 2023) with a batch size of 1, showing that even for large values of $|V_{\text{f-gram}}|$, embedding retrieval is not a bottleneck.

For a representative finding, in the 1B model variant, the baseline achieves an average perplexity of 16.082. Setting $|V_{\text{f-gram}}|$ to 10M improves the perplexity to 15.459 with

---

[2] https://github.com/allenai/OLMo/blob/v0.4.0/configs/official/OLMo-1B.yaml#L90

Table 2. Perplexity (lower is better) on the OLMo evaluation mixture. All models are trained for 200B tokens. SCONE consistently improves language modeling performance across all evaluation corpora. With 10M $V_{\text{f-gram}}$, a 1.3B model matches the performance of the 1.9B baseline. Similarly, with 1B $V_{\text{f-gram}}$, a 1B model matches the 1.9B baseline.

| Model size | c4-en | books | common-crawl | pes2o | reddit | stack | wiki | ice | m2de-s2orc | pile | wikitext-103 | Average |
|---|---|---|---|---|---|---|---|---|---|---|---|---|
| **1B baseline** | 16.813 | 21.570 | 16.752 | 11.682 | 22.612 | 3.360 | 14.453 | 15.281 | 27.900 | 10.429 | 16.053 | 16.082 |
| +10M $V_{\text{f-gram}}$ (0.6B $\mathcal{A}_{\text{f-gram}}$) | 16.087 | 20.963 | 16.039 | 11.270 | 21.797 | 3.274 | 13.777 | 14.979 | 26.361 | 10.128 | 15.371 | 15.459 |
| +10M $V_{\text{f-gram}}$ (1.8B $\mathcal{A}_{\text{f-gram}}$) | 15.727 | 20.429 | 15.473 | 11.124 | 21.388 | 3.231 | 13.454 | 14.709 | 25.785 | 9.956 | 15.104 | 15.125 |
| +1B $V_{\text{f-gram}}$ (0.6B $\mathcal{A}_{\text{f-gram}}$) | 15.846 | 20.593 | 15.684 | 11.071 | 21.411 | 3.213 | 13.543 | 14.702 | 26.026 | 9.889 | 15.077 | 15.187 |
| +1B $V_{\text{f-gram}}$ (1.8B $\mathcal{A}_{\text{f-gram}}$) | 15.158 | 19.680 | 14.857 | 10.761 | 20.757 | 3.133 | 12.964 | 14.220 | 24.958 | 9.553 | 14.354 | 14.581 |
| **1.3B baseline** | 15.994 | 20.157 | 15.921 | 11.148 | 21.634 | 3.248 | 13.721 | 14.651 | 26.583 | 9.927 | 15.143 | 15.284 |
| +10M $V_{\text{f-gram}}$ (0.6B $\mathcal{A}_{\text{f-gram}}$) | 15.509 | 19.816 | 15.407 | 10.887 | 21.022 | 3.192 | 13.260 | 14.372 | 25.450 | 9.757 | 14.616 | 14.844 |
| +10M $V_{\text{f-gram}}$ (1.8B $\mathcal{A}_{\text{f-gram}}$) | 15.193 | 19.587 | 14.995 | 10.795 | 20.735 | 3.171 | 13.071 | 14.272 | 25.258 | 9.674 | 14.438 | 14.654 |
| +1B $V_{\text{f-gram}}$ (0.6B $\mathcal{A}_{\text{f-gram}}$) | 15.270 | 19.510 | 15.106 | 10.707 | 20.763 | 3.139 | 13.073 | 14.177 | 25.009 | 9.546 | 14.397 | 14.609 |
| +1B $V_{\text{f-gram}}$ (1.8B $\mathcal{A}_{\text{f-gram}}$) | 14.803 | 18.996 | 14.541 | 10.502 | 20.296 | 3.085 | 12.637 | 13.971 | 24.533 | 9.357 | 13.971 | 14.245 |
| **1.9B baseline** | 15.270 | 19.017 | 15.184 | 10.719 | 20.752 | 3.163 | 13.119 | 14.095 | 25.461 | 9.570 | 14.229 | 14.598 |

a 0.6B $\mathcal{A}_{\text{f-gram}}$ model and to 15.125 with a 1.8B $\mathcal{A}_{\text{f-gram}}$ model, the later outperforming the 1.3B baseline (15.284). Increasing $|V_{\text{f-gram}}|$ to 1B further improves perplexity to 15.187 and 14.581 for the 0.6B and 1.8B $\mathcal{A}_{\text{f-gram}}$ models, respectively, surpassing the 1.9B baseline (14.598) despite requiring only half the inference-time FLOPS.

## 5. Related Work

**Contextualized Word Embeddings.** Words can have different meanings depending on context. Prior work has incorporated context into word embeddings, either from the entire sequence (McCann et al., 2017; Peters et al., 2018) or short $n$-grams (Gupta et al., 2019), before applying them to downstream tasks. Modern language models inherently use contextualized token embeddings, leveraging attention mechanisms. In this study, we extend the embedding layer to include contextualized f-gram embeddings for each token. A key novelty is that our approach allows embeddings to be precomputed and offloaded from accelerators, providing contextual embeddings for each token without increasing inference-time FLOPS.

**Scaling of Vocabulary Size.** Tao et al. (2024) show that larger models benefit from larger vocabularies, aligning with the trend of advanced language models whose vocabulary sizes have increased from a few dozen thousand (Devlin et al., 2019; Radford et al., 2019) to a few hundred thousand (Google, 2024; Adler et al., 2024; Dubey et al., 2024; DeepSeek, 2024). However, the optimal vocabulary size predicted by Tao et al. (2024) is still much smaller than the model size, e.g., a vocabulary size of 216K for a 70B parameter model. These findings motivate us to extend the embedding layer without changing the vocabulary size, fully exploiting the lookup nature of the input embedding layer.

**Tokenization in Language Models.** Our method assumes a predefined vocabulary from a trained tokenizer. Several popular algorithms exist for training tokenizers (Sennrich et al., 2016; Wu, 2016; Kudo, 2018b;a). In this work, we use a BPE tokenizer, following prior seminal works (Radford et al., 2019; Touvron et al., 2023). However, our method is not tied to any specific tokenization algorithm and can be applied seamlessly to others.

Tokenization-free language models have also been widely explored (Kim et al., 2016; Choe et al., 2019; Xue et al., 2022; Yu et al., 2023b; Wang et al., 2024; Deiseroth et al., 2024; Meta, 2024; Pagnoni et al., 2024). While we have not tested our method on tokenization-free models, we believe our core idea—introducing an off-accelerator embedding layer by precomputing embeddings for frequent input patterns—remains applicable.

Due to space limit, we discuss Mixture of Experts and Memory Layers, two established methods for scaling language models under a fixed FLOPS budget, in Appendix A.

## 6. Conclusion

We introduce SCONE, a scalable approach for generating $n$-gram contextualized embeddings for each input token without increasing inference-time FLOPS. These embeddings are learned during training and cached in off-accelerator storage for inference. SCONE enables two new aspects for scaling language models: (1) scaling the number of cached contextualized embeddings and (2) scaling the model size for learning them, both while maintaining fixed inference-time FLOPS. This is especially useful for latency-sensitive applications (Jones, 2021; Snell et al., 2024) and reducing serving costs. Future work could explore scaling embedding layers for other modalities; for instance, recent research underscores the importance of vocabulary size in visual modeling (Yu et al., 2023a).

## Impact Statement

This paper presents work whose goal is to advance the field of Machine Learning, and in particular, large language models. There are many potential societal consequences of our work, none which we feel must be specifically highlighted here.

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

---

**Algorithm 3** Basic Next-Word Prediction Model $M_{\mathcal{T},\mathcal{A},\mathcal{D}}$.

---

    **Parameters:**
- $\mathcal{T} : V_{\text{token}} \to \mathbb{R}^d$: token embedding layer,
- $\mathcal{A} : (\mathbb{R}^d)^{\leq N_{\max}} \to \mathbb{R}^d$: transformer model,
- $\mathcal{D} : \mathbb{R}^d \to \Delta_{V_{\text{token}}}$: prediction head.

    **Input:** $(\sigma_1, \ldots, \sigma_m) \in V_{\text{token}}^*$ for $m \leq N_{\max}$.
    **Output:** Probability distribution over next token $\hat{\sigma}_{m+1}$.
    **for** $i = 1, \ldots, m$ **do**
        $\boldsymbol{e}_i \leftarrow \mathcal{T}(\sigma_i)$ : Input embedding per token
    $\boldsymbol{e}_{\text{out}} \leftarrow \mathcal{A}(\boldsymbol{e}_1, \ldots, \boldsymbol{e}_m)$: Output embedding.
    **return** $D(\boldsymbol{e}_{\text{out}})$.

---

# A. Additional Related Work

Mixture of Experts (MoE) and Memory Layers are two well-established methods for scaling language models within a fixed FLOPS budget, as discussed below.

**Mixture of Experts**    MoE replaces traditional feedforward layers with parallel 'expert' layers, activating only one (or a few) per token via a lightweight router (Shazeer et al., 2017; Lepikhin et al., 2021; Fedus et al., 2022; Jiang et al., 2024; He, 2024). This allows scaling by increasing the number of experts without increasing active experts per token. However, all experts must reside on the accelerator, leading to higher memory usage.

**Memory Layers**    Memory layers store large sets of embeddings (continuous vectors) and retrieve nearest-neighbor embeddings during the forward pass via (approximate) similarity search (Weston et al., 2015; Sukhbaatar et al., 2015; Lample et al., 2019; Berges et al., 2024). These retrieved embeddings contribute to computations without adding to FLOPS. Despite advancements in similarity search (Lample et al., 2019; Johnson et al., 2019), memory layers still need to reside on accelerators, which increases memory demands to impractical levels at larger scales (Berges et al., 2024). Moreover, the embeddings in memory layers are typically updated through backpropagation, which also introduces sparse update challenges as memory scales.

Our approach focuses on input embedding layers, which we demonstrate can be efficiently offloaded from accelerators. This ensures constant memory usage and fixed FLOPS on the accelerator during inference. Additionally, by modifying only the input embedding layer, our method integrates seamlessly with both MoE and memory layer techniques.

# B. Additional Algorithms

---

**Algorithm 4** Next-word prediction with SCONE $M_{\mathcal{T},V_{\text{f-gram}},\mathcal{A}_{\text{f-gram}}|\mathcal{F},\mathcal{A}_{\text{main}},\mathcal{D}}$

---

    **Parameters:**
- $\mathcal{T} : V_{\text{token}} \to \mathbb{R}^d$: token embedding layer,
- $V_{\text{f-gram}} \subseteq V_{\text{token}}^{[2,n]}$: set of f-grams,
- **TRAINING**: f-gram transformer model
       ▷ $\mathcal{A}_{\text{f-gram}} : (\mathbb{R}^d)^{\leq n} \to \mathbb{R}^d$,
- **INFERENCE**: f-gram embedding layer
       ▷ $\mathcal{F} : V_{\text{f-gram}} \to \mathbb{R}^d$.
- $\mathcal{A}_{\text{main}} : (\mathbb{R}^d)^{\leq N_{\max}} \to \mathbb{R}^d$: main transformer model
- $\mathcal{D} : \mathbb{R}^d \to \Delta_{V_{\text{token}}}$: Prediction head.

    **Input:** $(\sigma_1, \ldots, \sigma_m) \in V_{\text{token}}^*$ for $m \leq N_{\max}$.
    **Output:** Probability distribution over next token $\hat{\sigma}_{m+1}$.
    $(\boldsymbol{e}_1, \ldots, \boldsymbol{e}_m) \leftarrow F_{\mathcal{T},V_{\text{f-gram}},\mathcal{A}_{\text{f-gram}}|\mathcal{F}}(\sigma_1, \ldots, \sigma_m)$ (Algorithm 2)
    $\boldsymbol{e}_{\text{out}} \leftarrow \mathcal{A}_{\text{main}}(\boldsymbol{e}_1, \ldots, \boldsymbol{e}_m)$
    **return** $D(\boldsymbol{e}_{\text{out}})$.

---

In Section 2, we discuss a simple next-word prediction model, $M_{\mathcal{T},\mathcal{A},\mathcal{D}}$, consisting of a token embedding layer $\mathcal{T}$, a

transformer model $\mathcal{A}$, and a prediction head $\mathcal{D}$. This model takes a token sequence $(\sigma_1, \ldots, \sigma_m)$, with each token from the token vocabulary $V_{\text{token}}$, and produces a probability distribution for the next token. We provide the pseudocode for $M_{\mathcal{T}, \mathcal{A}, \mathcal{D}}$ in Algorithm 3.

In Algorithm 2 in Section 3, we present the pseudocode for SCONE's process of generating contextualized f-gram embeddings. Next, we describe the end-to-end next-word prediction process using SCONE (Algorithm 4). Specifically, the process, denoted as $M_{\mathcal{T}, V_{\text{f-gram}}, \mathcal{A}_{\text{f-gram}} | \mathcal{F}, \mathcal{A}_{\text{main}}, \mathcal{D}}$, takes an input sequence $(\sigma_1, \ldots, \sigma_m) \in V_{\text{token}}^*$ and produces a distribution over the next token $\hat{\sigma}_{m+1}$. Note that in Algorithm 4, f-gram embeddings are generated with $\mathcal{A}_{\text{f-gram}}$ during training and retrieved from a lookup table $\mathcal{F}$ during inference.

## C. Challenges of Scaling Vocabulary Size in Embedding Layers

Scaling the vocabulary size is the most straightforward way to enlarge an embedding layer, but we find that larger vocabularies degrade performance beyond a certain threshold and significantly increase accelerator usage during decoding. We pre-train GPT-2 models (Radford et al., 2019) with three sizes of non-embedding parameters: 85M (small), 302M (medium), and 708M (large) on the WebText dataset (Peterson et al., 2019), testing six vocabulary sizes ranging from 32,768 to 2,097,152. The tokenizers are trained using the BPE algorithm (Gage, 1994; Sennrich et al., 2016). We follow the implementation in Tao et al. (2024), which allows token merges across word boundaries. Each model is trained on 80 billion tokens. Since larger vocabularies produce fewer tokens for the same dataset, they effectively enable models to process more data. Additional implementation details are provided in Appendix E.1.

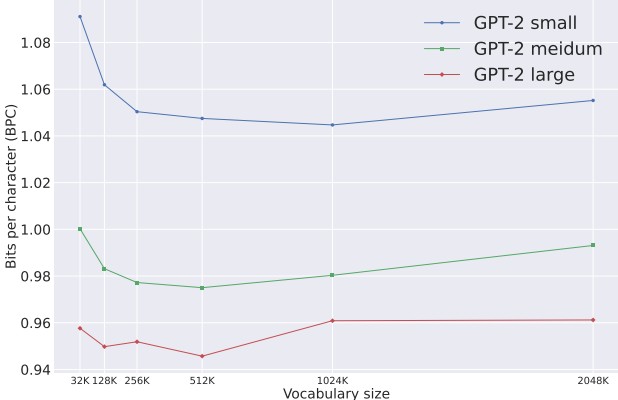

*Figure 8.* BPC of three model sizes on the validation set (lower is better). For all three model sizes, BPC initially improves as vocabulary size increases but eventually deteriorates.

Figure 8 presents the average bits per character (BPC) on the WebText validation set. We report BPC instead of cross-entropy loss because the latter is sensitive to vocabulary size, with larger vocabularies typically producing higher losses. BPC, by contrast, is a common vocabulary-insensitive metric for comparing models trained with different tokenizers (Huang et al., 2024). We observe that BPC for all three models initially improves with larger vocabulary sizes but eventually deteriorates.

Figure 9 shows the percentages of tokens that receive more than a given number of updates over 100 million training tokens. In standard embedding layers, gradients are directly backpropagated to the embedding vectors. With a fixed number of training tokens, larger vocabularies lead to fewer updates per token. For a vocabulary size of 2,097,152, only 7.3% of tokens receive more than 100 updates, compared to 97.6% for a vocabulary size of 32,768. This suggests that the performance drop for larger vocabularies may stem from sparse updates to per-token embedding vectors.

In addition to performance degradation, increasing the vocabulary size significantly raises accelerator usage during the inference stage. This is because predicting the next token involves running a linear layer and softmax operation across the entire vocabulary to identify the closest embedding. Figure 10 illustrates that both the number of embedding layer parameters stored on the GPU and the GPU memory cost increase linearly with vocabulary size. These costs are measured using a batch size of 1, a sequence length of 1024, and 16-bit precision.

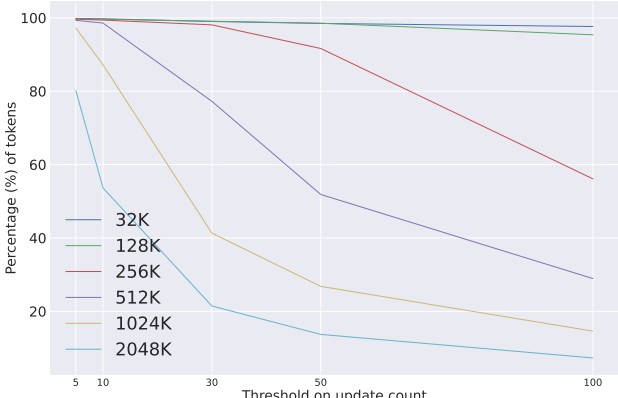

*Figure 9.* Percentages of tokens (y-axis) that receive more than a given number of updates (x-axis), measured over 100 million training tokens. As the vocabulary size increases, tokens receive increasingly sparse updates.

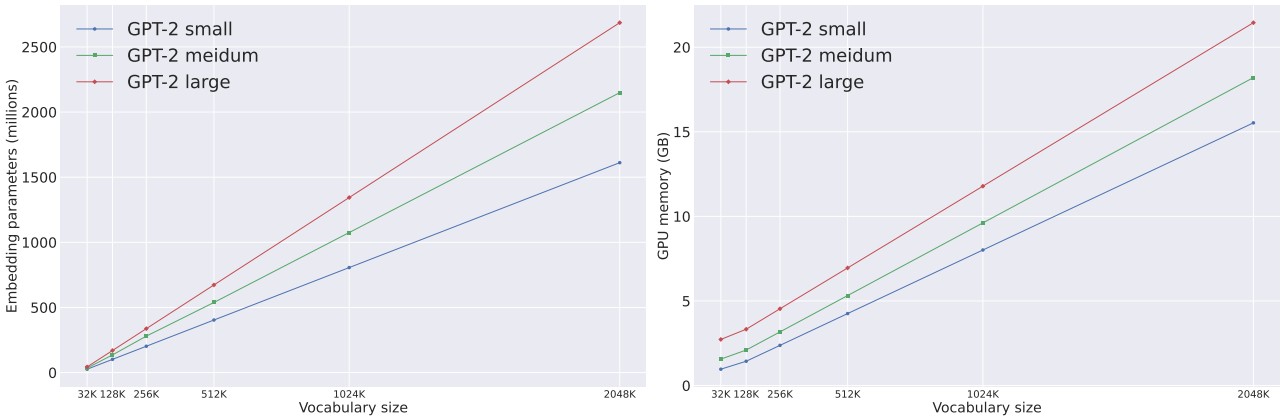

*Figure 10.* Number of embedding layer parameters on the GPU and corresponding GPU memory usage. Computational costs increase linearly with vocabulary size.

## D. Additional Experiments

### D.1. More Results for Training on WebText

**Varying Maximum f-gram Length.** In Section 4.1.1, we discuss the impact of varying the maximum f-gram length in $V_{\text{f-gram}}$ and present results on Wikitext-103. We observe that a relatively small maximum length is sufficient, as long as it is not too small, otherwise, the number of available $n$-grams for ranking becomes too limited. Here, in Figure 11, we show the corresponding results on WebText, which exhibit similar trends. The left $y$-axis represents the evaluation loss (averaged over three seeds), with the leftmost star indicating baseline performance. The right $y$-axis shows the average length of matched f-grams. As the maximum size increases, the loss initially decreases but then plateaus with some fluctuations. Meanwhile, the matched length rises initially before stabilizing for larger values.

**Varying $\mathcal{A}_{\text{f-gram}}$ Model Size.** In Section 4.1.3, we discuss the impact of varying the size of $\mathcal{A}_{\text{f-gram}}$ on evaluation perplexity for Wikitext-103. We find that increasing the model size leads to further performance improvements for a fixed $|V_{\text{f-gram}}|$. In Figure 12, we present the results on WebText, which show a similar trend. Model sizes in the legend correspond to inference-time sizes on accelerators. Dashed lines and stars on the left represent baseline performance. The evaluation perplexity improves as the size of $\mathcal{A}_{\text{f-gram}}$ grows.

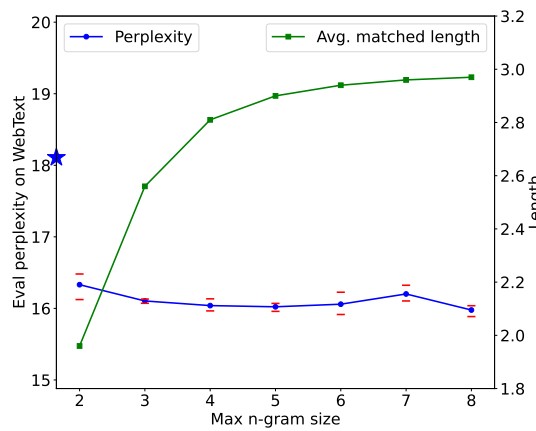

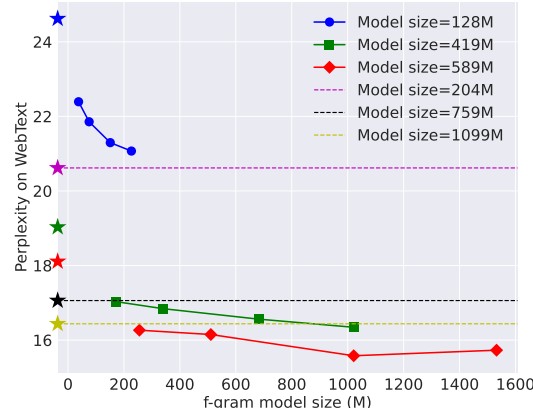

*Figure 11.* Effect of the maximum f-gram length in $V_{\text{f-gram}}$, evaluated on the WebText validation split.

*Figure 12.* Evaluation perplexity on WebText as a function of the size of $\mathcal{A}_{\text{f-gram}}$.

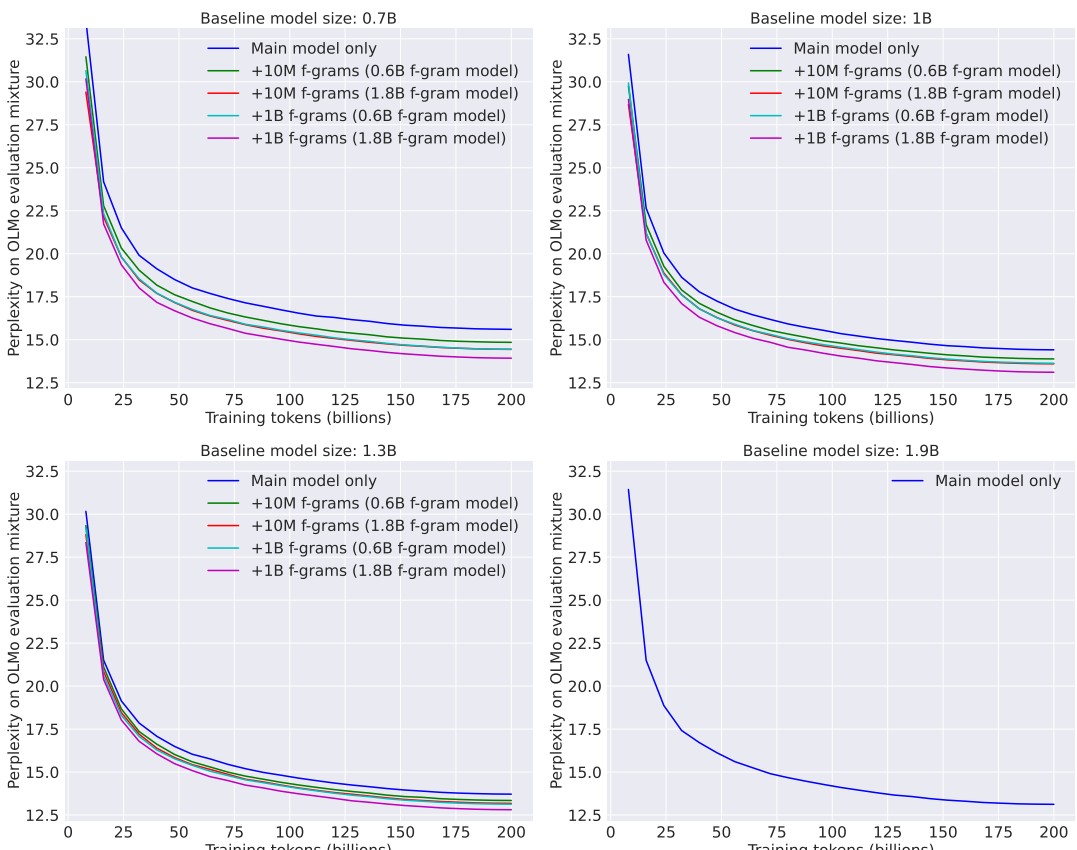

*Figure 13.* Average perplexity on the OLMo evaluation mixture throughout training. Models with SCONE enabled converge later, indicating stronger capacity, and achieve better perplexity.

## D.2. More Results for Training on OLMo Corpus

**Loss Curves over Training.** In Section 4.2, we pre-train four model variants with sizes of 0.7B, 1B, 1.3B, and 1.9B, evaluating SCONE on the first three. Each model is trained for 200B tokens, uniformly sampled from the OLMo-tokenized training corpus. The training token count is roughly 10 times more than the compute-optimal token count for a 1B model

suggested by Hoffmann et al. (2022), to ensure near-convergence. Figure 13 shows the evaluation loss throughout training. The loss curves indicate that models trained with SCONE converge later, suggesting that it effectively expands model capacity.

# E. Implementation Details

Here, we provide additional implementation details. We plan to open-source our code after the review process.

While f-gram lookup is efficient for inference, it creates a bottleneck during training since at training time transformer models process all token positions in parallel. This leads to GPU idle time when fetching the longest matching f-gram on the fly. To remove this bottleneck, after we construct the set of f-grams ($V_{\text{f-gram}}$), we pre-scan the training sequences to tag the longest matching length for each token. During training, we can then directly retrieve the corresponding f-gram for forward computation with the $\mathcal{A}_{\text{f-gram}}$ model.

For the $\mathcal{A}_{\text{f-gram}}$ model, we use an absolute position embedding layer where the maximum position equals the longest $n$-gram in $V_{\text{f-gram}}$. Within each batch, all f-grams are padded to the longest $n$-gram length in that batch. We train all models with the bfloat16 precision.

## E.1. WebText

| Parameters (million) | d_model | ffw_size | n_layers |
|---|---|---|---|
| 128 | 1024 | 4096 | 6 |
| 204 | 1024 | 4096 | 12 |
| 491 | 1536 | 6144 | 12 |
| 759 | 1536 | 6144 | 24 |
| 589 | 1536 | 6144 | 18 |
| 1099 | 1536 | 6144 | 36 |

*Table 3.* Baseline model configurations for pre-training on WebText. For constructing the f-gram model ($\mathcal{A}_{\text{f-gram}}$), we vary the number of layers in the 128M, 491M, and 589M variants and discard the token embedding layer.

For pre-training on WebText (Peterson et al., 2019), we follow Radford et al. (2019) and set the batch size and sequence length to 512 and 1024, respectively. Radford et al. (2019) do not specify the number of training tokens or optimizer details. We train the models for 80B tokens, roughly doubling the count in Radford et al. (2018). For optimization, we use AdamW (Loshchilov, 2017) with a weight decay of 0.1. Following Hoffmann et al. (2022), we set the maximum learning rate to $2 \times 10^{-4}$ and apply a cosine learning rate scheduler. We list the model configurations in Table 3.

## E.2. OLMo Tokenized Training Corpus

| Parameters (million) | d_model | ffw_size | n_layers |
|---|---|---|---|
| 711 | 2048 | 8192 | 12 |
| 1014 | 2048 | 8192 | 18 |
| 1316 | 2048 | 8192 | 24 |
| 1920 | 2048 | 8192 | 36 |

*Table 4.* Baseline model configurations for pre-training on OLMo corpus. For constructing the f-gram model ($\mathcal{A}_{\text{f-gram}}$), we use the 711M and 1920M configurations and discard the token embedding layers.

For pre-training on the OLMo tokenized training corpus, we follow the optimizer settings for the 1B variant in Groeneveld et al. (2024) [3]. All models use a sequence length of 2048 and are trained on 200B tokens from sequences uniformly sampled from the corpus. We use DeepSpeed (DeepSpeed, 2024) with ZeRO stage 1 to reduce GPU memory usage. ZeRO stage 1 partitions the optimizer state across GPUs. Our hardware supports training models up to a training-time size of 3B parameters. We list the model configurations in Table 4.

[3] https://github.com/allenai/OLMo/blob/v0.4.0/configs/official/OLMo-1B.yaml#L40

