# OpenReview forum: "Scaling Embedding Layers in Language Models"
_ICML.cc/2025/Conference — Submitted to ICML 2025_

### Official Review · Reviewer_EL23 · 2025-02-21

**Overall Recommendation:** 4

**Summary:**

This paper proposes SCONE, which is an extended n-gram embedded layer to improve model's performance. SCONE introduces contextualized embeddings for frequently-used n-grams. While these embeddings are learned from a small Transformer model, they can also be precomputed and stored to avoid additional latency. The experiment shows that SCONE achieves better performance without additional inference cost.

## Update after rebuttal
My score keeps unchanged since I have no misunderstandings with the authors.

**Claims And Evidence:**

This paper makes systematic experiments on the inference cost and enhanced performance.

**Essential References Not Discussed:**

All essential references are discussed.

**Experimental Designs Or Analyses:**

The experimental designs are reasonable and the results are solid.

**Methods And Evaluation Criteria:**

This paper leverages well-accepted evaluation criteria.

**Other Comments Or Suggestions:**

No other comments.

**Other Strengths And Weaknesses:**

I think the novelty of this paper is worth praising. There is no clear weaknesses in this paper.

**Questions For Authors:**

No other questions.

**Relation To Broader Scientific Literature:**

This is an interesting and novel research topic. Most of research on LLM architecture is usually on the Attention and FFN Part. There are only a few research focusing on the simple input embedding layer. However, this paper is relatively a new research direction to enhance model's capability without additional inference overhead.

**Theoretical Claims:**

No theoretical claims.

---

> ### Author Rebuttal · Authors · 2025-04-01
>
> Thank you for your supporting review! We appreciate your recognition of the novelty and contributions of our work. We also welcome any further comments you may have.

---

### Official Review · Reviewer_LBJ6 · 2025-03-15

**Overall Recommendation:** 4

**Summary:**

The paper proposes a new method, SCONE, to expand the embedding layer. Instead of directly expanding the vocabulary size, which usually leads to the sparsity issue (long-tailed symbols/tokens receive sparse updates due to their data sparsity), the paper chose to expand the embedding layer by incorporate frequent n-grams (f-grams)embeddings for given inputs. The n-grams embeddings are precomputed by a separate model and can be offloaded to CPU and secondary memory during inference. This allows scaling the modelling capacity of embedding layer without incurring additional inference cost. Experiments on GPT-2 tokenizers and OLMo architecture demonstrate the proposed method has potential to scale either the number of cached n-grams or the model learning them without increasing inference costs.

**Claims And Evidence:**

The claims are well-supported:
- Solid experiments across multiple model sizes (128M to 1.3B) and datasets
- Clear ablation studies on key parameters
- Convincing measurements of storage and latency impacts
- Strong results showing a 1B model with SCONE outperforming a 1.9B baseline

Some are not clear to me. But I feel they are more or less deserving deeper studies rather than this work which focuses on empirical speedup in inference time.
- The paper seems to assume expand embedding layer with f-grams can enhance model performance. But it doesn't thoroughly validate that additional n-grams are necessarily useful across all contexts. I feel in some context, using pure token embeddings should be enough for the task.
- The approach assumes n-grams only be added in embedding layers, but self-attention and feedforward layers also implicitly store bi-gram and tri-gram patterns. Directly offloading these computations could avoid repeated processing and yield further efficiency gains.

**Essential References Not Discussed:**

SCONE's approach to embedding expansion could benefit from situating itself within a broader context of research across multiple domains. The following references would provide important theoretical grounding and suggest potential applications beyond the paper's current scope:

- Recent work [1-4] points out that some implicit circuits or paths inside transformers might already capture n-grams. These works are relevant to understand how SCONE achieves the speedup theoretically. For example, is the offloading of frequent n-grams replacing some of the original n-gram pathways inside transformer (for example, jet bi-gram path or jet n-gram path in [4])? This represents a tradeoff between the in-context computation and the lookup table computation.

- Frequency-related performance degradation is not uncommon in general embedding-based models. The issue of sparsity when scaling vocabulary size is not unique to LLMs but also exists in other embedding systems. Solutions can include adaptively regularizing the embeddings or sparsifying them [5,6] in the recommender systems. Actually, SCONE might be more powerful in the domain of recommender models where embeddings take up an even larger portion of the entire model, which can be a better scenario for SCONE (as in LLMs it seems that scaling the model sizes might diminish the benefit of SCONE).

- Expanding vocabulary and scaling embedding layers is quite important for multilingual applications. Some work on expanding vocabulary [7,8] focuses on expanding embeddings for this purpose. It could be worthwhile to discuss how SCONE can help these domains. For example, instead of finding n-grams in English, can we find and embed the n-grams in a target language? This would show that SCONE can be an effective way to do cross-lingual transfer without incurring non-emebdding inference cost.

By incorporating these references and exploring connections to these adjacent fields, SCONE could establish stronger theoretical foundations while simultaneously demonstrating broader applications beyond language modeling, potentially strengthening both its academic contribution and practical impact.

[1] [Neurons in Large Language Models: Dead, N-gram, Positional](https://arxiv.org/abs/2309.04827)

[2] [Transformer Feed-Forward Layers Are Key-Value Memories](https://aclanthology.org/2021.emnlp-main.446/) (EMNLP 2021)

[3] [Transformer Feed-Forward Layers Build Predictions by Promoting Concepts in the Vocabulary Space](https://aclanthology.org/2022.emnlp-main.3/) (EMNLP 2022)

[4] [Jet Expansions of Residual Computation](https://arxiv.org/pdf/2410.06024)

[5] [λopt Learn to Regularize Recommender Models in Finer Levels](https://arxiv.org/pdf/1905.11596) (KDD 2019)

[6] [Learnable Embedding Sizes for Recommender Systems](https://arxiv.org/pdf/2101.07577) (ICLR 2021)

[7] [On the Cross-lingual Transferability of Monolingual Representations](https://aclanthology.org/2020.acl-main.421.pdf) (ACL 2020)

[8] [Improving Language Plasticity via Pretraining with Active Forgetting](https://arxiv.org/abs/2307.01163) (NeurIPS 2023)

**Experimental Designs Or Analyses:**

yes, they make sense
- Consistent training across baselines and variants
- Good ablation studies
- Appropriate scaling from small to large models
- Testing across diverse datasets

**Methods And Evaluation Criteria:**

yes, they make sense
- Their f-gram selection process builds on established BPE approaches
- They use standard metrics (perplexity) on common benchmarks
- They measure both model performance and practical deployment factors

**Other Comments Or Suggestions:**

some quotations are single commas rather than double commas

**Other Strengths And Weaknesses:**

#### Strengths
- The motivation for scaling embeddings is compelling and addresses a real bottleneck in language model efficiency
- The approach of replacing partial model computation with precomputed frequent n-gram embeddings is innovative and practical
- The method is conceptually simple and does not rely on particular architectural design of the model. Thus it has great generalizability and could potentially be used for improving performance in other domains beyond language modeling, particularly in recommender systems and cross-lingual transfer.

#### Weaknesses
I would not say these are weakness but rather like interesting directions to explore in the future.
- The focus on embedding offloading is somewhat narrow - the paper could explore offloading other non-embedding compute to CPU as well, such as implementing a query-answer datastore for frequent queries that could similarly speed up inference
- While the paper demonstrates good performance gains, it's not entirely clear whether SCONE can outperform alternative methods like Mixture of Experts (MoE) when focusing solely on inference speedup, though the authors do discuss MoE in the appendix

**Questions For Authors:**

**Q1:** Can you explain in simple terms how the speedup is achieved? The paper states: "We demonstrate that a 1B parameter model with SCONE outperforms a baseline model requiring ∼2× more inference-time FLOPs."

**Q2:** Does SCONE impact sample efficiency during training? Or does the introduced f-gram embedding require more data to train effectively?

**Q3:** Counting n-grams on a large corpus is computationally expensive and can take days. Could you estimate the time complexity of your counting algorithm and report the runtime for computing n-grams over a typical pretraining corpus?

**Q4:** Looking at Figure 6, SCONE appears to work well with smaller models but seems to bring less improvement to perplexity as model size increases. Could you explain why the benefit margin diminishes with larger models?

**Relation To Broader Scientific Literature:**

The authors position their work in relation to three main areas of prior research:

- Contextualized Word Embeddings: They acknowledge previous work on incorporating context into word embeddings and highlight that their key novelty is allowing embeddings to be precomputed and offloaded without increasing inference-time FLOPS.
- Scaling of Vocabulary Size: Recent findings from Tao et al. (2024) show that larger models benefit from larger vocabularies but note that optimal vocabulary sizes are still much smaller than model sizes. This motivates the proposed approach of extending the embedding layer without changing the vocabulary size.
- Tokenization in Language Models: They discuss how their method assumes a predefined vocabulary but is not tied to any specific tokenization algorithm, making it compatible with various approaches.
- They also discussed MoE and memory layers in the appendix

**Theoretical Claims:**

no theoretical claims made though it could be interesting to figure out how much additional expressiveness is added with the f-gram embeddings and how they tradeoff the non-embedding computation with embeddings.

---

> ### Author Rebuttal · Authors · 2025-04-01
>
> Thank you for your review. Please find our responses below, we’re happy to discuss further if needed. We’ve also included downstream evaluations in our response to Reviewer iJdC.
>
> **1. Missing references**
>
> We thank the reviewer for the insightful references. We have incorporated all the references in the following discussions, and added them to Related Work (Sec 5).
>
> “**Implicit $n$-gram patterns in transformers.** Recent work analyzing the internal mechanisms of transformers has shown that these models often utilize implicit $n$-gram patterns for prediction (Geva et al., 2021; Geva et al., 2022; Voita et al., 2023). For instance, Chen et al., 2024 show that certain attention heads detect specific $n$-gram patterns, while MLPs can perform linguistic operations such as adding the “-ing” suffix. These findings underscore the importance of $n$-gram information in language modeling and offer a potential explanation for the effectiveness of SCONE. An interesting future direction is to examine how SCONE's f-gram embeddings interact with the transformer’s implicit $n$-gram patterns.”
>
> “**Embedding sparsity in multilingual applications and recommender systems.** This work focuses on a common setting for training LLMs: language modeling on large-scale text corpora, primarily in English. However, scaling embedding layers presents challenges beyond this context, particularly due to frequency-related performance degradation caused by sparsity. Multilingual applications are one such scenario. Two phrases in different languages may refer to the same concept but correspond to different embeddings. Their embeddings should ideally be close. Recent work explores methods for learning transferable embeddings in cross-lingual settings (Artetxe et al., 2020; Chen et al., 2023). Another relevant example is scaling the embeddings for recommender systems (Chen et al., 2019; Liu et al., 2021), where embeddings often dominate the model's parameter count due to the high cardinality of user or item categories. For both scenarios, SCONE’s strategy, i.e., parameterizing large embedding tables using a neural network, provides a complementary approach to mitigate sparsity issues.”
>
> **2. The focus on embedding offloading is somewhat narrow.**
>
> We agree that exploring offloading beyond input embeddings could further reduce inference costs and our work can be viewed as a first step. A key research challenge along this direction is deciding what should serve as keys in such a system. We have added the following discussion to the last section:
>
> “An interesting future direction is to extend SCONE beyond short $n$-grams to include longer and frequent queries. A key challenge would be designing effective keys for such queries. Using raw text as keys may lead to low hit rates, as semantically similar queries often differ at the surface level. Alternatively, using semantic embeddings as keys would require discretization methods to map continuous embeddings to a set of keys that supports efficient indexing.”
>
> **3. Whether SCONE can outperform methods like MoE.**
>
> Comparing with MoE is indeed an interesting future work. That said, SCONE offers a key advantage: while both aim to scale model capacity under fixed inference FLOPS, MoE requires all expert weights to reside on accelerators, since any token might activate any expert. In contrast, SCONE is designed for **fixed accelerators memory usage at inference**.
>
> **4. Explain in simple terms how the speedup is achieved. (1B with SCONE v.s. ~2x inference FLOPS baseline)**
>
> The 1B SCONE-enabled model outperforms the 1.9B baseline. The 1.9B baseline requires ~2x inference FLOPS. Thank you for pointing this out; we’ve clarified this in the introduction.
>
> **5. Training sample efficiency of SCONE?**
>
> No, SCONE does not reduce sample efficiency. All models are trained on the same number of tokens, and as shown in Figure 13, the improvements are consistent throughout training.
>
> **6. Complexity of the counting algorithm & its runtime.**
>
> As noted in Section 3.1, there is an efficient implementation for the counting algorithm that requires $n-1$ linear passes over the corpus ($n$ is the max f-gram length). On 1T tokens using 8 processes and $n=5$, processing took about 10 hours. We've included this in the revision.
>
> **7. SCONE appears to bring less improvement to perplexity as model size increases.**
>
> As perplexity decreases, achieving the same absolute reduction becomes more difficult due to the nature of language modeling. In Figure 6, we used a linear scale on the y-axis, which can make improvements at lower perplexity appear smaller. We’ve uploaded a new figure (at [this anonymous link](https://github.com/Anonymous-figure-upload/scone_author_response)) that uses a log y-axis (following recent scaling law studies, e.g., Figure 2 in Hoffmann et al., 2022). The improvements in log scale appear more consistent.
>
> **8. Some quotations are single commas.**
>
> We have changed the single comma quotations in line 567.

---

> > ### Comment · Reviewer_LBJ6 · 2025-04-04
> >
> > Thanks for addressing my concerns and sharing the new evaluation results including the new figure on perplexity.  Below are my comments.
> >
> > Given the large portion of embeddings in large language models ( as we expand to multiple languages, the vocabulary size will lead to a even larger portion of parameters being the embeddings), the topic of this paper is quite relevant and timely, receiving a large audience as this method can reduce the inference cost significantly.
> >
> > Evaluation-wise: The authors demonstrate good results using not only perplexity but also metrics on common benchmarks during rebuttal. This new result further convinces me that the paper deserves a clear accept.
> >
> > Writing and presentation-wise: My other concerns was mainly among how the work can be well contextualized regarding the literature of scaling embedding-based models. The authors provided a revised related work section while also sufficiently discussing their method compared to methods like MoE during the rebuttal stage. I would suggest the authors add a discussion over the scope (bullet point 2 and 3 along with some of the related works) in the appendix.
> >
> > N-grams-wise: N-grams are interesting themselves as both linguistic analysis targets and the basic component for language modeling. Although the early n-grams models are not as competent as the latest transformer architectures, they can be quite a useful tool for further scaling up the models [9]. Additionally, it is very likely that the LLMs themselves are good n-gram approximators as shown by the line of work on interpretability and theory of transfoemer [1-4,10-12].  I also believe the good work in this line is coming rather than a bold statement like "n-grams has been given up in LLM era".
> >
> > Overall, I believe this paper deserves a clear accept given its relevance, novelty and good emprical results.
> >
> > [9] Infini-gram: Scaling Unbounded n-gram Language Models to a Trillion Tokens https://arxiv.org/abs/2401.17377
> >
> > [10] A Mathematical Framework for Transformer Circuits https://transformer-circuits.pub/2021/framework/index.html
> >
> > [11] Can Transformers Learn n-gram Language Models? https://arxiv.org/abs/2410.03001
> >
> > [12] The Role of n-gram Smoothing in the Age of Neural Networks https://arxiv.org/abs/2403.17240

---

> > > ### Author Response · Authors · 2025-04-04
> > >
> > > We thank the reviewer for the follow-up comments and appreciate the helpful suggestions.
> > >
> > > * Regarding the suggestion to add a discussion over the scope.
> > >
> > > We will extend the discussion of related work and promising future directions as suggested. This will include: (1) offloading embeddings for longer sequences beyond $n$-grams, (2) connections with implicit $n$-gram circuits in transformers, (3) potential applications of SCONE in multilingual and recommender system settings, and (4) the connections and distinctions among our approach, MoE, and memory layers.

---

### Official Review · Reviewer_AdYi · 2025-03-20

**Overall Recommendation:** 1

**Summary:**

This paper introduces a technique for expanding input embedding layers to improve the performance of language models. The experimental results show that the solution mentioned in this paper outperforms a 1.9B parameter baseline.

**Claims And Evidence:**

I think the writing of this paper can be improved. Maybe all the claims made in the submission are supported by convincing evidence, but the authors must rewrite some parts of the paper. For example, the caption of Figure 1 (top) claims that with 10M f-grams, the 1.3B model matches the 1.9B baseline, while with 1B f-grams, the 1B model surpasses it. However, the legend of Figure 1 (top) is "+10M f-grams (0.6B f-gram model)", "+10M f-grams (1.8B f-gram model)", "+1B f-grams (0.6B f-gram model)", and "+1B f-grams (1.8B f-gram model)". It seems that no line in Figure 1 represents the 1.3B model.

**Essential References Not Discussed:**

I think this paper almost includes all related works.

**Experimental Designs Or Analyses:**

I think this paper should include more evaluation.

**Methods And Evaluation Criteria:**

Almost no existing open-sourced model uses the same architecture as that developed in this paper.

**Other Comments Or Suggestions:**

Here are my comments:
1. lines 88-93, the authors claim "Our experiments with GPT-2 models (Radford et al., 2019) pre-trained on WebText (Peterson et al., 2019)
confirm these limitations: Only 7.3% of embedding vectors in a 2M vocabulary receive more than 100 updates over 100M training tokens, compared to 97.6% for a 32K vocabulary". However, it is known to all that the 7B model is always trained by 1T tokens (https://arxiv.org/pdf/2302.13971), therefore, it is not clear why embeddings receive very few updates when we train the 7B model.
2. Lines 391 and line 396, I feel confused that +10M Vf-gram (0.6B Af-gram) has different performance over c4-en, books, etc. The caption of Table 2 mentions that all models are trained for 200B tokens. In addition, where is the 1.3B model mentioned in the caption of Table 2?
3. The Perplexity of OLMo is important. However, what people really care about is the accuracy of models over downstream tasks. Could you evaluate your pre-trained model over downstream tasks such as Arc-Challenge, Arc-Easy, and Hellaswag?
4. The concept shares similarities with RAG in certain respects. However, according to Table 1, deploying this model requires substantial memory or disk space. Therefore, the model with f-gram embedding is hard to deploy at an edge device.
5. Given the vocabulary size of Gemma is 256,000, I'm concerned that the memory and disk requirements for the f-gram embedding layers for the Gemma model will surge significantly.

The writing issue:
1. the x-label of Figure 1 should be inference FLOPs, not inference-time FLOPs.
2. Is it possible to keep the line width consistent?
3. I guess the red line in Figure 5 is an error bar. However, the error bar always contains a vertical line.
4. As for Figure 6, the legend overlaps with the line.

**Other Strengths And Weaknesses:**

Strengths:
1. This paper proposes a solution to expand the embedding layer.
2. The experimental results show the effectiveness and efficiency of the developed approach.

Weaknesses:
1. The manuscript's quality is lacking. Major parts should be rewritten, and the figures need to be polished.
2. The motivation of this paper is not clear. Especially, this architecture is not utilized by any open-weighted model. In addition, n-gram has been given up in the LLM era. I'm unsure why the authors employ n-grams to enhance the performance of large language models.

**Questions For Authors:**

See comments and suggestions.

**Relation To Broader Scientific Literature:**

This paper considers f-gram embeddings for transformer-based models.

**Theoretical Claims:**

This paper does not include any proofs for theoretical claims.

---

> ### Author Rebuttal · Authors · 2025-04-01
>
> We thank the reviewer for the suggestions. Please find our response below.
>
> **1. The manuscript's quality is lacking.**
>
> We have carefully addressed the concerns regarding writing (see below). Please see the updated Figure 1 (and its caption), Figure 5, and Figure 6 at [this anonymous link](https://github.com/Anonymous-figure-upload/scone_author_response). We would like to clarify that these issues pertain to presentation and do not affect the validity of our main claims.
>
> **1.1** Figure 1: the line of 1.3B model and the x-label.
>
> The 1.3B model corresponds to the column with inference FLOPS of $6.20\times 10^{12}$. We have revised the caption to improve clarity.
>
> We have changed the x-label, and other occurrences of “inference-time” FLOPS in the paper, to inference FLOPS.
>
> **1.2** Line width not consistent.
> We have adjusted the width of all figures to be $0.9\times$ the text column width.
>
> **1.3** Figure 5: error bars.
> We have updated Figure 5 to include vertical lines in the error bars.
>
> **1.4** Figure 6: legend overlaps with the line.
> We have adjusted the figure to eliminate any overlap between the legend and the line.
>
> **2. The motivation is not clear. This architecture is not utilized by any open-weighted model. In addition, n-gram has been given up in the LLM era.**
>
> Our motivation is to reduce inference FLOPS by trading computation for RAM or SSD, which are significantly cheaper and more abundant than accelerators. A natural target for this is the embedding layer, due to its inherent lookup-based structure. While we agree that the NLP community has largely moved away from traditional n-gram methods, recent work showed that $n$-grams still play important roles in transformers (due to space limit, please see the references in reviewer LBJ6’s review). We hope our work offers a new perspective on how n-grams can be leveraged to improve the efficiency of LLMs.
>
> **3. … it is known to all that the 7B model is always trained by 1T tokens**
>
> In lines 88–93, our intention is to highlight that increasing the vocabulary size can eventually degrade model performance. We use the update counts over 100M tokens to help explain this phenomenon. We have revised the manuscript to clarify this point:
>
> “In Appendix C, we train GPT-2 models for 80B tokens with vocabulary sizes ranging from 32K to 2M, and observe performance degradation as the vocabulary size exceeds 512K. This degradation may be attributed to the increasing sparsity of updates per token as the vocabulary grows. …”
>
> With 1T training tokens, the absolute number of updates each embedding receives will increase. However, the relative sparsity remains: larger vocabularies still result in fewer updates per embedding on average.
>
> **4. Lines 391 and line 396, I feel confused that +10M Vf-gram (0.6B Af-gram) has different performance over c4-en, books, etc. … In addition, where is the 1.3B model mentioned in the caption of Table 2?**
>
> Although lines 391 and 396 use the same f-gram configuration, they correspond to different main model sizes. Specifically, the results from lines 390 to 394 are based on the 1B model, while lines 395 to 399 correspond to the 1.3B model.
>
> To clarify this distinction, we have added the following to the caption of Table 2:
>
> “We train three baseline models of sizes 1B, 1.3B, and 1.9B. For the 1B and 1.3B baseline models, we apply our SCONE method with four different configurations, and present the results directly below each corresponding baseline model.”
>
> **5. Could you evaluate your pre-trained model over downstream tasks?**
>
> Yes, we evaluated the zero-shot performance of our models on PIQA, MMLU, HellaSwag, ARC-Easy, ARC-Challenge, and CommonsenseQA, following [the recent implementation in the OLMO codebase](https://github.com/allenai/OLMo/blob/main/olmo/eval/downstream.py). Due to space constraints, please see **our response to reviewer iJdC** for the full results. The downstream evaluation outcomes align with the perplexity trends.
>
> **6. … deploying this model requires substantial memory or disk space.**
>
> The configurations in Table 1 use large f-gram embedding layers to demonstrate that even the most resource-intensive setup explored in the paper remains feasible in certain server-based settings.
>
> As shown in Figure 6, SCONE offers clear improvements even with much smaller embedding sizes. For example, with a 512K f-gram embedding layer—approximately 20× smaller than the 10M setting in Table 1—the perplexity of the 589M base model improves from 18.1 to 16.8 on WebText.
>
> **7. The memory and disk requirements for the Gemma model will surge significantly.**
>
> For a given embedding dimension, the storage usage of the f-gram embedding layers is only determined by the number of f-grams, which is a configurable parameter independent of the vocabulary size. Therefore, please note that a larger vocabulary does not lead to higher memory or disk requirements compared to smaller vocabularies.

---

> > ### Comment · Reviewer_AdYi · 2025-04-02
> >
> > Thanks for your rebuttal! I reviewed the author's rebuttal and the comments from other reviewers.
> >
> > However, I have the following concerns:
> > 1. I agree that the methodology mentioned in this paper can improve the accuracy of models over downstream tasks in some cases since this paper proposes a novel embedding/tokenizer solution. However, the improvement is limited from the results of 1.9B (56.75%) and 1B + 1B f-grams (57%).
> > 2. I think the presentation of this paper should be improved. For instance, the authors should polish the figures and add some examples (as a response to **Reviewer iJdC**) to aid readers' understanding. This paper appears to be poorly prepared.
> > 3. Since you offload the f-gram embedding layer to the storage, I do not think inference FLOPs is still a good metric. The users really care about end-to-end latency. Inference FLOPs are not equal to end-to-end latency.
> > 4. Almost all papers shared by **Reviewer LBJ6** are before 2024 (only one paper was submitted in 2024), therefore, I still believe that n-gram is not an important technique in the LLM era.
> >
> > In conclusion, I would not be surprised if this paper is accepted as a poster. However, if this paper is rejected by ICML, I recommend that the authors enhance the presentation of the paper before the next submission. Honestly, I consider this paper to be borderline. Thanks!

---

> > > ### Author Response · Authors · 2025-04-02
> > >
> > > Thank you for your follow-up comments! For improving clarity and presentation, we’ll make sure [the polished figures](https://github.com/Anonymous-figure-upload/scone_author_response) are included in the revision. We will also include the example in the response to Reviewer iJdC.
> > >
> > > Regarding the comment on inference FLOPs vs. end-to-end latency, we respectfully note that we did report end-to-end token generation speed at the bottom of Figure 1 (i.e. counting the latency of loading f-gram embeddings to GPU). The plot used the [vLLM package](https://github.com/vllm-project/vllm). The figure shows that: **1)** when f-gram embeddings are stored in CPU memory, the impact on latency is negligible,  **2)** when they are stored on NVMe, the generation speed decreases by ~20% (e.g., for the 1B model the speed reduces from ~250 tokens per second to ~200 tokens per second).
> > >
> > > We appreciate your thoughtful feedback and hope our clarifications support a potential adjustment of the score toward a borderline recommendation.

---

### Official Review · Reviewer_iJdC · 2025-03-22

**Overall Recommendation:** 3

**Summary:**

The paper presents a new method for scaling the vocabulary of LLMs. Given some base vocabulary, a set of frequent n-grams is calculated. When such a n-gram is seen, a small transformer (called an f-gram model) is applied to the n-gram embeddings to produce a new embedding. This embedding is then fed to the larger LLM in lieu of the original n-gram. During evaluation, the f-gram embeddings can be precomputed and cached in a separate table. The authors evaluate this method experimentally and find:

1. It improves evaluation perplexity.
2. Perplexity improves as a function of the number of embeddings.
3. Perplexity improves as a function of the f-gram model size.

**Claims And Evidence:**

The claims are generally supported by evidence, but I'm not convinced that perplexity is the best evaluation metric. It will change depending on the tokenizer, and thus it's not a good apples-to-apples metric. It would be better to use MMLU/hellaswag etc.

**Essential References Not Discussed:**

Na

**Experimental Designs Or Analyses:**

The design of experiments is sensible, but the evaluation metric is not good imo.

**Methods And Evaluation Criteria:**

No, see above.

**Other Comments Or Suggestions:**

na

**Other Strengths And Weaknesses:**

Pros:

1. The idea is simple and natural.
2. The writing is clear.
3. There are a lot of good experiments.

Cons:

1. The evaluation metric (perplexity) is not a good metric. Because of this, I do not think that the evaluation is good. So I'm not convinced that the method works in practice.

**Questions For Authors:**

1. Can you add numbers for standard LLM evals like MMLU, hellaswag and so on?
2. Can you clarify how the system works during inference? If I decode 3 tokens from the base vocabulary and find that they match an n-gram from the f-gram model, should the model recompute the k,v-cache for these three tokens with the embeddings from the matched n-gram?

EDIT: I have increased my score in response to the non-perplexity eval numbers.

**Relation To Broader Scientific Literature:**

Na

**Theoretical Claims:**

na

---

> ### Author Rebuttal · Authors · 2025-04-01
>
> We thank the reviewer for the comments. Please find our response below.
>
> **1. The evaluation metric (perplexity) is not a good metric. Can you add numbers for standard LLM evals like MMLU, hellaswag and so on?**
>
> While perplexity is a commonly used metric for evaluating language models, we acknowledge that additional downstream evaluations can further strengthen our work. In response, we have incorporated zero-shot evaluations on standard benchmarks including PIQA, MMLU, HellaSwag, ARC-Easy, ARC-Challenge, and CommonsenseQA, following [the recent implementation in the OLMO codebase](https://github.com/allenai/OLMo/blob/main/olmo/eval/downstream.py). The results are shown below. Thank you for suggesting; we will add the benchmark results in the revision.
>
> | Model  | PIQA    | HellaSwag | ARC-Easy | ARC-Challenge | Commonsense QA | MMLU_var | Avg    |
> |--------|---------|-----------|----------|---------------|----------------|----------|--------|
> | 1B baseline | 73.57 | 60.93   | 69.47  |    31.76    |   48.73      | 37.61  | 53.67 |
> | 1.9B baseline | 75.31 |  65.86  | 74.21  |   36.78     |      49.71   |  38.64 | 56.75  |
> | 1B + 10M f-grams|  73.95 | 63.58   | 70.35  |  32.09      |   49.96      |  39.25 | 54.86 |
> | 1.3B + 10M f-grams| 75.04 |  65.52  | 75.26  |   36.44     |   49.96      | 38.54  | 56.79 |
> | 1B + 1B f-grams| 75.31  |  67.05  |  72.45 |     36.44   |   50.78      |  39.97 | 57.00 |
>
>
> Applying SCONE, i.e., adding f-gram embeddings, does not increase inference FLOPS and requires only off-accelerator storage. Notably, the 1.9B baseline incurs roughly twice the inference FLOPS of the 1B model. The downstream evaluation results align with the perplexity trends and further reinforce our main claims.
>
> **2. Can you clarify how the system works during inference? If I decode 3 tokens from the base vocabulary and find that they match an n-gram from the f-gram model, should the model recompute the k,v-cache for these three tokens with the embeddings from the matched n-gram?**
>
> To clarify, SCONE does not require recomputing the (k,v) cache during inference. Instead, it fetches the f-gram embedding for each decoded token individually. For example, if the current context is [t0, t1] and the newly decoded token is t2, and we find a match for [t0, t1, t2] in the f-gram embedding layer, we use the f-gram embedding of [t0, t1, t2] as the input embedding for t2 in the main model. This f-gram embedding of [t0, t1, t2] corresponds to the embedding of the last token embedding in the output of the f-gram model (precomputed for [t0, t1, t2]) and only involves simple lookups during inference. If it helps, we can add this example in the revision.

---

### Decision · Program_Chairs · 2025-05-01

**Decision:**

Reject

**Comment:**

The paper proposed a novel method to scale the number of parameters in the input embedding layer. Given a set of vocabulary and a corpus, a set of frequent n-grams are obtained as extra vocabulary. Each n-gram is assigned with an embedding vector to enhance the input embedding for LM. Experiment results show that this method improves model performance for small scale models without increasing the parameter count that involved in compute.

While the paper provided relatively novel path for scaling language model. The experiments are limited to smaller scales (less than 2B). It's unclear whether it will help some more practical model size, such as larger than 7B. It would be good to see if the proposed method could be used to improve existing larger language model.